# Recombinant Pseudorabies Virus Usage in Vaccine Development against Swine Infectious Disease

**DOI:** 10.3390/v15020370

**Published:** 2023-01-28

**Authors:** Mo Zhou, Muhammad Abid, Shinuo Cao, Shanyuan Zhu

**Affiliations:** 1Jiangsu Key Laboratory for High-Tech Research and Development of Veterinary Biopharmaceuticals, Engineering Technology Research Center for Modern Animal Science and Novel Veterinary Pharmaceutic Development, Jiangsu Agri-Animal Husbandry Vocational College, Taizhou 225306, China; 2Viral Oncogenesis Group, The Pirbright Institute, Ash Road Pirbright, Woking, Surrey GU24 0NF, UK

**Keywords:** pseudorabies virus, virus modification, virus-vectored vaccines, recombinant PRV, swine infectious disease

## Abstract

Pseudorabies virus (PRV) is the pathogen of pseudorabies (PR), which belongs to the alpha herpesvirus subfamily with a double stranded DNA genome encoding approximately 70 proteins. PRV has many non-essential regions for replication, has a strong capacity to accommodate foreign genes, and more areas for genetic modification. PRV is an ideal vaccine vector, and multivalent live virus-vectored vaccines can be developed using the gene-deleted PRV. The immune system continues to be stimulated by the gene-deleted PRVs and maintain a long immunity lasting more than 4 months. Here, we provide a brief overview of the biology of PRV, recombinant PRV construction methodology, the technology platform for efficiently constructing recombinant PRV, and the applications of recombinant PRV in vaccine development. This review summarizes the latest information on PRV usage in vaccine development against swine infectious diseases, and it offers novel perspectives for advancing preventive medicine through vaccinology.

## 1. Introduction

Pseudorabies virus (PRV) is the etiological agent of pseudorabies (PR), belonging to the *Herpesviridae* family, primarily affects pigs and can occasionally be transmitted to cattle, goats, sheep, cats, and dogs [1,2,3,4,5]. The PRV virus is a double-stranded DNA virus approximately 143 kb in size, consisting of unique long (UL), internal repeat short (IRS), unique short (US), and a terminal repeat short (TRS) [6,7,8]. There is a minimum of 70 open reading frames (ORFs) in the genome encoding 70–100 viral proteins, including virulence-related proteins and replicase, as well as many proteins that are not essential for PRV replication [9,10]. The large PRV genome allows the insertion of several kilobases (kb) foreign DNA sequences. The expression of foreign genes does not influence the virus replication [11]. Therefore, multivalent vaccines were developed using PRV as a vector, in which the major antigen of different swine pathogens was expressed [12]. 

Live vector vaccine can stimulate humoral immunity and solid cellular immunity, which is the main goal for developing a swine infectious disease vaccine. Scientists have generated PRV vaccine strains with few side effects and good immunity by inserting foreign genes or knocking out self-virulence genes. This review summarizes the biological characteristics of PRV, the methodological principle and technology platform for constructing characteristics of PRV, the methodological principle and technology platform for constructing PRV recombinants efficiently, and the applications of PRV recombinants in vaccine development. This review is intended to serve as a reference for PRV live vector vaccine research and development. 

## 2. The Biological Characteristics of PRV

There are 10 glycoproteins encoded by PRV, which are divided into two groups (essential or non-essential glycoproteins) according to the degree to which viral replication depends on them [13,14]. The glycoprotein B (gB) plays a crucial role in membrane fusion during infection and in spreading viruses from cell to cell [15,16,17]. A major glycoprotein encoded by the glycoprotein C gene mediates the attachment of PRV to target cells through heparin-binding domains [11,18,19,20]. In susceptible animals, the protective immune responses were induced by gB, gC, and glycoprotein D (gD) [21,22,23]. In the viral envelope of infected cells, a heterodimer complex formed by glycoprotein I (gI) and glycoprotein E (gE) plays a crucial role in PR infection and transmission [24,25,26,27]. 

Bartha-K61, the original PRV vaccine strain, contains the deletion of US2, gE, gI, and US9 genes. This vaccine strain has been widely used to control PR worldwide. Since the 1990s, few swine PR cases have been reported due to PRV Bartha-K61 [4,28,29,30,31]. However, the PR outbreak occurred at the farm where pigs were vaccinated with Bartha-K61, which was caused by PRV variants and resulted in severe economic losses from 2011 [32,33,34,35,36,37,38]. Vaccines remain the most effective method to control PRV infection. Consequently, the PRV variant has been used to generate several gene mutant vaccines with gE, gI, and TK deletions. When challenged with PRV variants, these mutants provided adequate protection [39,40,41,42,43,44,45,46,47,48]. By introducing foreign genes derived from other porcine pathogens into the PRV variants and subsequent vaccination, it is possible to establish immune protection against these swine infectious diseases.

## 3. PRV Recombinants Construction

There are some non-essential regions in the large genome of PRV. As a result of this character, several kb of foreign DNA can be accommodated by PRV without compromising its stability. Reverse genetics plays a crucial role in PRV gene modification, and with the development of molecular biology techniques, many reverse genetic techniques have been used to construct the recombinant PRV.

### 3.1. Homologous Recombination

In the past, PRV recombinants were generated by homologous recombination (HR) in permissive cells. The construction of recombinant viruses based on the principle of HR can be divided into two steps: construction of a transfer plasmid and screening of the recombinant virus. Usually, the virus genome is used as a template to amplify the two gene fragments used as recombinant homologous arms by PCR technology, and then the foreign gene expression cassette is inserted between the homologous arms to construct the transfer vector containing the foreign gene expression cassette and the left and right homologous arms. Gene deletion or replacement is achieved by homologous recombination of the transfer vector and parent strain in eukaryotic susceptible cells. At present, there are two methods used for transfection. One method is co-transfection of the transfer vector and the parent virus genome, and another method is transfection of the transfer vector first and then inoculation of the virus. The reporter gene carried on the transfer vector is used to screen the plaque of the recombinant virus. Generally, the recombinant virus with infective activity can be obtained after 5–10 generations of screening. By using this method, Tong et al. and Wang et al., constructed a recombinant PRV strain expressing the E2 protein of CSFV in a gE-deleted PRV variant strain [49,50]; Yan et al. constructed the recombinant virus with additional expression of the gC gene [51]; Zheng et al. produced recombinant PRV that expressed VP2 protein of porcine parvovirus and porcine IL-6 [52]. Because of the low efficiency of HR, it is not commonly used in the construction of PRV recombinants.

### 3.2. Bacterial Artificial Chromosome

In comparison with homologous recombination, bacterial artificial chromosomes (BAC) are more efficient. Red/ET recombination technology can be used to modify BAC in *Escherichia coli*, which can be divided into two steps [53]. The first step is to construct an expression cassette with marker genes and both ends containing homologous arm sequences and transform them into competent cells containing viral BAC by electroporation. After resistance screening, correct recombinants were obtained. The second step was constructing an expression cassette with foreign genes and both ends containing homologous arm sequences. The recombinants with deleted marker genes were obtained by counter-resistance screening. Zhang et al. construct a TK/gE-deleted AH02LA BAC using the virulent PRV AH02LA strain and inserted the pig epidemic diarrhea virus (PEDV) variant spike (S) expression cassette in four different noncoding regions using the En Passant mutagenesis method [54]. Furthermore, to develop the recombinant vaccine against the PRV variants, Zhang et al. constructed a BAC clone of Bartha-K61. Using Bartha-K61 BAC as the backbone, the gD and gC of Bartha were replaced with that of PRV variants (AH02LA) via the En Passant method. When compared to the parental Bartha-K61, there was no difference in growth properties in the gD/gC-substituted Bartha-K61 ST cells [22].

### 3.3. Fosmid Library

The traditional HR method is not efficient in obtaining recombinant viruses. Creating recombinant BAC constructs is a labor-intensive and time-consuming process. When the whole genome PRV is cloned to the BAC plasmid, fragment deletion often occurs during the electroporation process, affecting recombination efficiency. In contrast, fosmid library generation is more efficient. The huge genome of PRV is randomly sheared into various fragments, then each segment is ligated with a fosmid vector to screen out the various combinations covering the whole genome of PRV, and the fosmid combination was co-transfected into virus-susceptible eukaryotic cells to rescue the virus [55,56]. Construction of recombinant virus only needs to modify the corresponding fosmid in *E coli.* Since the fosmid only contains part of the PRV genome, the efficiency of obtaining virus mutants is greatly improved. Modification of fosmid in *E. coli* can also utilize Red/ET recombination technology. A fosmid library based on the PRV-TJ strain was constructed by Zhou et al., and recombinant PRV was rescued by transfecting a group of fosmid directly into Vero cells. Furthermore, the green fluorescent protein-expressing reporter virus was rescued successfully by the Red/ET system [55]. Abid et al. constructed a PRV co-expressing E2 of classical swine fever virus (CSFV) and Cap of porcine circovirus type 2 (PCV2) by using this fosmid library platform [57]. In addition, Qi et al. constructed a fosmid library for the PRV-SC strain. The EGFP gene was inserted into the N-terminal of the UL 36 gene of PRV-SC via the Red/ET recombination [56]. 

### 3.4. CRISPR/Cas9

Molecular biotechnology has dramatically contributed to studying viruses’ replication and vaccines. The CRISPR /Cas9-mediated genome editing system has been widely applied as a powerful tool to construct PRV mutants. The researchers used CRISPR/Cas9 technology to construct PRV gE, gI, or TK gene-deleted recombinants [42,43,46]. Furthermore, CRISPR/Cas9 can be used in combination with HR and BAC technology to improve the efficiency of recombination [53,58]. Fu et al., used CRISPR/Cas9 gene-editing technology to construct a double reporter virus with stable expression of EGFP and firefly luciferase [59]. Feng et al. used this technology to insert the African swine fever virus (ASFV) CD2v gene into the TK, gE, and gI-deleted PRV Fa strain. The ASFV CD2v was stability expressed in the recombinant virus [12]. Wu et al. inserted the tandem repeats of the ORF2 gene into PRV gE and gG sites through CRISPR/Cas9 mediated HR and constructed a bivalent vaccine based on the PRV-HNX strain [60]. To optimize the method used for generating infectious recombinant PRV expressing predicted or known ASFV immunogenic proteins, Fuchs et al. developed an efficient strategy for the insertion of foreign genes in PRV vector based on marker-enforced recombination and CRISPR/Cas9. They co-transfected the defective PRV Bartha-K61 BAC DNA (possesses a deletion of glycoprotein G and an additional deletion encompassing the initiation codon and promoter of glycoprotein D, which plays an essential role in binding to receptors), the rescue transfer plasmid (A PRV-encoded transgene flanked by homologous recombination sequences and an intact gD promoter are included in this construct), and the CRISPR/Cas9 plasmid to the cell. Transgene substitutions were observed in 99% of the obtained progeny viruses [61]. Additionally, CRISPR/Cas9 and Cre/Lox greatly improved multi-gene editing efficiency and reduced vaccine development time. Yao et al. combined the Cre/Lox and CRISPR/Cas9-mediated gene insertion system, deleted the TK and gE gene, and inserted the immunoregulatory factor gene simultaneously [62]. 

Homologous recombination is the traditional method to generate the recombinant PRV. BAC was used later and allowed manipulation of the whole genome in *Escherichia coli*. The homologous recombination method is labor-intensive due to several rounds of plaque purification. The BAC system was more efficient than homologous recombination, but the generation of recombinant BAC construct is time-consuming, and the huge PRV genome is easily broken. The fosmid library and CRISPR/Cas9-based genetic manipulation platform for PRV offer several advantages over the conventional technology, but some disadvantages still appeared during the construction of the recombinants (Table 1). Therefore, we need to consider the advantages of various approaches and develop a more efficient way to construct the recombinant PRV. 

## 4. Principle behind Recombinant PRV

The expression of foreign genes is an important index to evaluate the efficacy of the live recombinant vaccine, which is affected by the promoter and the insertion site of the foreign gene. 

### 4.1. Promoter Affects Foreign Gene Expression

The initial transcription of foreign genes is the critical step of gene expression, and the rate of transcription initiation is the rate-limiting step of gene expression. The promoters and related regulatory sequences are the premises of constructing a good expression system. The regulation elements of exogenous gene expression include a promoter, enhancer, Kozak sequence, and stop codon. The expression level of the foreign gene in PRV mainly depends on the strength of the upstream promoter, and a generally strong promoter is selected. The gG and gE promoters were always chosen to construct recombinant PRV, which have unique structures and vigorous activity [57,60]. A more potent heterologous enhancer or promoter, such as the HCMV or MCMV promoter, was also evaluated instead of the gG promoter [50,54]. Different promoters were tested to express the ASFV gene in PRV, such as HCMV, MCMV, and a chimeric promoter (CAG) consisting of the HCMV enhancer and the chicken beta-actin promoter [61]. Compared with cytomegalovirus immediate-early promoters, the CAG promoter enhanced the expression of foreign genes in PRV. 

### 4.2. Insertion Site Is the Main Factor, Affects the Expression of Foreign Genes 

For a live vector vaccine strain, genetic stability is crucial. The PRV genome contains numerous sites for inserting and expressing heterologous genes, but almost all are located near the gI/gE, TK, or gG genes. In a study by Tong et al., the gD and gG genes were inserted with E2 expression cassettes, and insertion of the cassettes did not significantly affect in vitro replication of the recombinant virus. And the piglets challenged with virulent PRV are protected by recombinant viruses [50]. Wu et al. also inserted the tandem repeats of the PCV2 Cap protein gene into PRV gG and gE sites. A high level of PCV2 Cap protein expression was detected in this recombinant PRV [60]. Intergenic regions between gG and gD proved to be excellent sites for expressing foreign genes, according to these results. To identify suitable areas to insert foreign genes into PRV genomes, Zhang et al. inserted an S protein expression cassette of a PEDV variant in different noncoding regions of the PRV genome with deletion of TK, gE, and gI, such as US2-1, UL11-10, UL46-27, and UL35-36. The US2-1 area is critical for PRV replication, as the insertion of a foreign gene failed to rescue the virus, another three noncoding regions are suitable sites for the insertion of the S gene; S gene mRNA expression was higher when it was inserted in the location of UL11-10 [50,54]. Therefore, the UL11-10 is a new suitable insertion site for S gene expression.

So far, when selecting the insertion sites of foreign genes, researchers have considered the non-essential regions of virus replication first. Although the genome sequence of PRV contains a large number of non-essential regions for replication, the basic theories of the relationship between these regions and virus virulence, immune escape mechanism, or host range need to be further studied. At the same time, the expression of exogenous genes in the vector is affected by the size of the exogenous fragment, the promoter of the expression vector, and the insertion site effect caused by the insertion of exogenous genes. Therefore, a certain size of a foreign gene, stable insertion sites of a foreign gene, and safe and efficient expression elements are the basis for the application of recombinant live vector vaccines. 

## 5. Applications of Recombinant PRV 

The large genome of PRV contains numerous insertion sites that allow heterologous genes to be integrated and expressed. Consequently, PRV has developed into a powerful vector system for expressing foreign proteins. 

### 5.1. PRV Bartha K61 Strain as a Vector for Expressing Exogenous Antigens

In pigs, PRV Bartha K61 replicates reliably and elicits a wide range of humoral and cellular immune responses. Additionally, in order to establish clinical protection against different pathogens, foreign antigens can be introduced into the Bartha K61 backbone and vaccinated. By introducing the GP5 gene from the porcine reproductive and respiratory syndrome virus (PRRSV) into Bartha K61, and subsequently vaccinating, significant clinical protection was achieved, and pathological lesions were reduced after the PRRSV challenge in piglets [63]. Further, mice were protected against virulent challenges with swine influenza H3N2 by vaccination with a Bartha vector that expressed the hemagglutinin of swine influenza H3N2 [64]. Furthermore, pigs vaccinated with Bartha vectors expressing the pandemic H1N1 swine-origin influenza virus hemagglutinin or neuraminidase also demonstrated significant inhibition of virus replication after a challenge with the H1N1 virus [65]. Due to the limitations of Bartha K61 in protecting against PRV variants, Zhang et al. constructed the gD/gC-substituted Bartha K61, in which the gD/gC of Bartha K61 was substituted with the gD/gC of PRV variants (AH02LA). The gD/gC-substituted Bartha K61 was safe, and it effectively protected against virulent PRV variants [22]. Therefore, Bartha K61 was a safe and effective backbone for multivalent vaccine development.

### 5.2. Attenuated PRV Variant as a Vector for Expressing Exogenous Antigens

Since 2011, farms immunized with Bartha-K61 have experienced PR outbreaks caused by PRV variants. Therefore, the researchers try to generate several gene mutant vaccines involving gE/TK, gI/gE, and gI/gE/TK deletion based on different PRV variants. These vaccines provided adequate protection against the PRV variant challenge. These gene mutant vaccines also are used as a vector system for foreign proteins to develop multivalent live virus-vectored vaccines against several swine infectious diseases. Tong et al. and Wang et al. constructed a recombinant expressing the CSFV E2 protein and evaluated its efficacy against both CSFV and PRV variant strains [49,50]. The recombinant PRV appears to be a promising recombinant vaccine candidate for the control and eradication of PRV variants and CSFV. Through the fosmid library platform, Abid et al. constructed a recombinant co-expressing E2 of CSFV and Cap of PCV2 with gE/gI/TK gene deletions [57]. The recombinant strain was safe for pigs and rabbits, and anti-PRV antibodies could be detected in the serum of immunized rabbits and pigs. But anti- E2 and PCV2 antibodies could not be detected, which was presumed to be because the expression level of E2 and Cap proteins are too low to induce antibody production. In addition, a tandem repeat of PCV2 Cap protein gene ORF2 that links with a protein quantitation rationing linker was constructed by Wu et al., and endogenous PRV promoters were used to drive PCV2 Cap2 expression in PRV [60]. High levels of neutralizing antibodies were detected 14 days after immunization of the recombinant bivalent vaccine candidate, which was maintained at days 21, 42, and 60. In wild boars and domestic pigs, the African swine fever virus (ASFV) causes a fatal disease, and there is no effective vaccine against the ASFV. In order to develop the live vector vaccine against ASFV. Feng et al. insert the CD2v into the PRV variant Fa stain (TK, gE, and gI deleted) and obtained a recombinant strain expressing CD2v gene [12]. The recombinant virus stimulated the production of anti-CD2v antibodies and specific cellular immune responses. Furthermore, the recombinant virus could protect mice against the virulent strain (PRV-Fa) infection.

### 5.3. Attenuated PRV Variant as a Vector for Expressing Immunoregulatory Factors 

Lymphokine-activated killer cells recognize infected cells associated with PRV gC during cell-mediated immunity, and PRV gC plays a role in cytotoxic T-lymphocytes recognition of infected cells [22,66,67]. To enhance gC expression, Yan et al. inserted an additional expression cassette of the gC gene into the gI and gE deletion regions [51]. Additional gC gene expression enhanced the efficacy of the PRV gI/gE-deleted vaccine in pigs. A hematopoietic growth factor, Fms-related tyrosine kinase 3 ligand (Flt3L), plays an important role in hematopoietic cell differentiation. In the periphery, FLT3L and its receptor regulate homeostatic DC division [68,69]. An FLT3L-dependent pathway mediates vaccine-induced protective immunity and enhances the activation of T cells [70]. Based on the immunomodulatory functions of Flt3L, a TK/gE gene deleted and Flt3L co-expressed recombinant PRV was constructed by Yao et al. [62]. The recombinant gene showed potential function in DC activation and protective immune response enhancement. These results indicated that t Flt3L is an ideal adjuvant, and the PRV can load the immunoregulatory factors and achieve simultaneous immunization with the adjuvant and vaccine. 

### 5.4. Attenuated PRV Variant as Vector for Expressing Reporter Genes 

Up to now, the mechanism of PRV variant virulence enhancement is unknown. A reporter virus is a valuable tool for basic virology studies. Therefore, Fu et al. constructed a double reporter virus with stable expression of EGFP and firefly luciferase based on the PRV variant [59]. The recombinant virus showed similar biological characteristics to the parental strain, including a stable viral titer and luciferase activity throughout 20 passages. Furthermore, Zhou and Qi constructed the reporter virus based on the PRV variant and classical stain, respectively [55,56]. The EGFP was fused into the amino-terminal of UL36 in these recombinants. The single viral particles with green fluorescence were used to monitor retrograde and anterograde moving virions in the axon, these reported viruses will accelerate the understanding of the biology of the PRV variant.

## 6. Conclusions and Prospects

The genetically engineered live vector vaccine can effectively avoid the traditional vaccines’ defects, simplify the breeding industry’s immunization procedures, and improve the efficiency of epidemic prevention and control. Attenuated PRV-based vector vaccines do not induce any clinical signs of PR, but provide complete protection against PRV and other swine pathogens, and it has been widely used in the study of a recombinant multivalent vaccine against swine infectious disease. The BAC technology, CRISPR/Cas9 technology, and fosmid library were used to construct PRV recombinant, significantly improving the efficiency of exogenous gene expression (Table 2). But we still need to consider how to ensure the genetic stability of the immune genes, and how to ensure the safety of recombinant viruses during live vector vaccine development. The significant virulence genes were deleted during the development of the recombinant PRV live vector vaccine, significantly reducing its virulence to animals, However, the safety of the recombinants still needs to be analyzed in detail. There are extensive replication non-essential regions in the PRV genome, which provide a basis for optimizing recombinant sites. The selection of better insertion sites and safer and more efficient expression elements, especially an efficient promoter, is the basis of recombinant live vector vaccine application. Further research and utilization of the advantages of recombinant PRV live vector vaccine will play an essential role in the prevention and control of animal diseases. The recombinant herpes virus live vector vaccine will play an important role in animal disease prevention and control in the future.

## Figures and Tables

**Table 1 viruses-15-00370-t001:** Advantages and disadvantages of recombinant PRV construction approaches.

Approaches	Advantages	Disadvantages
Homologous recombination (HR)	The procedure is relatively easy; Site-specific insertion of foreign genes.	Construction of transfer vector is time-consuming and labor-intensive; The efficiency of homologous recombination is low; The purification of viruses is time-consuming.
Bacterial Artificial Chromosome (BAC)	More efficient than homologous recombination; Cloning the entire genome into a plasmid facilitates genome modification.	Generation of recombinant BAC construct is time-consuming; The PRV genome is easily broken, and once the genome is broken, infectivity is lost and a recombinant virus cannot be obtained.
Fosmid library	Generation of the fosmid library is more efficient; High structural stability.	The fracted genome was inserted into 5-6 fosmids; 5-6 fosmids need to be co-transfected.
CRISPR/Cas9 system	Highly efficient; Simultaneous targeting of multiple sites.	It is necessary to construct homologous arms; Off-target effects usually occurred; This method also requires the purification of the virus.

**Table 2 viruses-15-00370-t002:** The characteristic features of recombinant PRV.

Vector	Exogenous Genes	Insertion Sites	Promoters	Modification of PRV	Immunization Dose/Route	Animal Model	Efficacy	Reference
Bartha-K61	respiratory syndrome virus (PRRSV) GP5	UL23 (TK) site	CMV promoter	homologous recombination (HR)	10 ^7.0^ pla- que-forming unit (PFU)/ intranasally (i.n.) and intramuscularly (i.m.)	4-week-old piglet	confer significant protection against clinical disease and reduce pathogenic lesions induced by PRRSV challenge in vaccinated pigs	[63]
Bartha-K61	hemagglutinin (HA) gene of swine influenza virus (SIV)	not mentioned	SV40 promoter	HR	10 ^5.0^ PFU/i.n.	8-week-old mice	protect mice from heterologous virulent challenge	[64]
Bartha-K61	HA of swine-origin H1N1 virus.	gG gene locus	MCMV promoter	bacterial artificial chromosome (BAC) technology-mediated HR	2×10 ^7.0^ PFU/i.n.	7-week-old pigs	protected pig from clinical signs after challenge with a related swine-origin H1N1 influenza A virus	[65]
Bartha-K61	gD and gC genes of the AH02LA strain	gD and gC gene locus		BAC technology and HR	10 ^6.0^ TCID _50_ / i.m	4-week-old piglet	PRV B-gD&gC S is safe for piglets, and provides complete clinical protection against a pseudorabies variant (AH02LA) challenge	[22]
Bartha-K61	open reading frames E199L, CP204L (p30) and KP177R (p22) of African swine fever virus.	gG gene locus	CAG promoters	BAC technology and CRISPR/Cas9				[61]
PRV variant (HN1201strain)	enhanced green fluorescent protein (EGFP) and firefly luciferase	between gE partial and gI partial	EGFP expression was under the control of the CAG promoter (a synthetic promoter composed of a CMV enhancer and chicken β-actin promoter); the firefly luciferase expression cassette was under the control of the SV40 promoter	CRISPR/Cas9				[59]
PRV variant (AH strain)	glycoprotein C	between gD and US9	CMV promoter	HR	10 ^7.0^ TCID _50_ / i.m (piglet); 10 ^5.0^ TCID _50_ / i.m (mice)	4-week-old piglet; 4-week- old SPF Kunming mice	additional insertion of gC gene could enhance the protective efficacy in PRV gI/gE-deleted vaccine in pigs	[51]
PRV variant (HNX strain)	cytokine Fms-related tyrosine kinase 3 ligand (Flt3L)	after gD	gD promoter	CRISPR/Cas9 and Cre/Lox systems	10 ^5.0^ TCID _50_ / i.m	Six-week-old female BALB/c mice	Flt3L can activate DCs and enhance protective immune responses of recombinant pseudorabies virus with TK/gE gene deletion	[62]
PRV variant (TJ strain)	classical swine fever virus (CSFV) E2 glycoprotein and capsid (Cap) protein of porcine circovirus type 2 (PCV2)	E2 expression cassette was inserted after US9; Cap was fused with gG and co-expressed with gG	E2 expression was under the control of CMV promoter; Cap was under the control of the gG promoter	fosmid library platform and Red/ET systems	10 ^7^, 10 ^6^, and 10 ^5^ TCID _50_ / i.m (rabit); 10 ^6.0^ TCID _50_ / i.m (pig)	6-week-old rabbits; 6-week-old pigs	rPRVTJ-delgE/gI/TK-E2-Cap elicited detectable anti-PRV antibodies, but not anti-PCV2 or anti-CSFV antibodies	[57]
PRV variant (AH02LA strain)	S gene of a Porcine epidemic diarrhea virus (PEDV) variant	UL11-10, UL35-36, UL46-27	MCMV promoter	BAC technology and En Passant method				[54]
PRV variant (JS-2012)	CSFV E2 glycoprotein	between the gG and gD genes	not mentioned	HR	10 ^5.0^ TCID _50_ / i.m	3-week-old piglets	induce the production of Abs to the gE protein of PRV or to the CSFV proteins other than E2	[50]
PRV variant (TJ strain)	CSFV E2 glycoprotein	between gE partial and gI partial	CMV promoter	HR	10 ^4^, 10 ^5^, and 10^6^ TCID _50_ / i.m	6-week-old piglets	provided complete protection against the lethal challenge with either the PRV TJ strain or the CSFV Shimen strain	[49]
PRV variant (HNX strain)	tandem repeats of porcine circovirus type 2 (PCV2) Capprotein gene ORF2	gE and gG sites		CRISPR/Cas9 and HR	10 ^5.8^ TCID _50_/ i.m	4-week-old female BALB/c mice	high titer of specific antibodies for PRV and neutralized antibodies for PCV2 were detected	[60]
PRV (Fa strain)	CD2v of African swine fever virus	PRV UL23 (TK) site	CMV promoter	CRISPR/Cas9 and HR	10 ^5.0^ TCID _50_ / i.m	5-week-old SPF mice (ICR)	PRV-ΔgE/ΔgI/ΔTK-(CD2v) recombinant strain has strong immunogenicity	[12]
PRV	P12A and 3C of Foot-and-mouth disease virus (FMDV)	between gE partial and gI partial	CMV promoter	HR	10 ^6.0^ TCID _50_ / i.m	6-week-old large white piglets	PRV-P12A3C induced a high level of neutralizing antibody and FMDV-specific lymphocytes.	[71]
PRV variant (TJ strain)	EGFP	fused with UL35	UL35 promoter	fosmid library platform and Red/ET				[55]
PRV (SC strain)	EGFP	fused with UL36	UL36 promoter	fosmid library platform and Red/ET				[56]

## Data Availability

The data presented in this study are available in the insert article.

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
