# Peer review of "Recombinant Pseudorabies Virus Usage in Vaccine Development against Swine Infectious Disease"

_viruses, 2023, doi:10.3390/v15020370_

Round 1

Reviewer 1 Report

The review from Zhou and colleagues has focused on the latest research progress about the usage of pseudorabies virus-vector vaccine against the swine infectious disease. The authors have referred to recent scientific reports and the paper is well-written. I have the following comments.

1.      Abstract, line 18, PRV can maintain a long immunity, is there any literature documenting how long the immunity can be maintained? Please be specific.

2.      In line 35, the authors claimed the PRV genome allows the insertion of sizeable foreign DNA sequences. Is there any data on how big the foreign gene can be accommodated into the PRV genome while it doesn’t influent the virus replication? Please be specific.

3.      It’s better to make a table to summarize the application of different strategies of rPRV generation and compare the advantages and disadvantages of these strategies.

4.      For the principle behind the recombinant PRV. Whether the size of the foreign gene would impact the genome stability? Whether the insertion sites and size of the foreign gene should be considered together to construct the rPRV? If so, please discuss the relationship among the insertion site, the size of the foreign gene, and the genome stability.

5.      Table 1 provides a good summary of the PRV-based vaccine. To provide a compressive overview of vaccines, the authors could consider including the immunization dose, route, animal model, and efficacy of these vaccines, such as immunogenicity and protectivity.

6. Line 243, there is a grammar mistake.

Author Response

We are grateful to the editor and reviewers for their critical comments and valuable suggestions, which have helped us to improve our manuscript. We have revised the manuscript, and would like to re-submit it for your consideration. As outlined in the following responses, we have taken all these comments and suggestions into account in the revised manuscript. The new text is highlighted in red color in the revised manuscript. We point by point responses to the referee’s comments are listed below. We hope that the revised manuscript is acceptable for publication.

Responses to the comments of Reviwer 1

The review from Zhou and colleagues has focused on the latest research progress about the usage of pseudorabies virus-vector vaccine against the swine infectious disease. The authors have referred to recent scientific reports and the paper is well-written. I have the following comments.

Response:

We sincerely thank the reviewer for his/her constructive and positive comments, which have helped us to improve our manuscript. As outlined in the following responses, we have taken these comments and suggestions into account in the revised manuscript.

  1. 1. Abstract, line 18, PRV can maintain a long immunity, is there any literature documenting how long the immunity can be maintained? Please be specific.

Response:

We thank the reviewer for this comment. We have checked some reference, and some reports indicated that, PRV antibody remain at a high level until 130 days after the first immunization in piglets and pregnant sows. So we have emphasized this point in the abstract.

  1. In line 35, the authors claimed the PRV genome allows the Insertion of sizeable foreign DNA sequences. Is there any data on how big the foreign gene can be accommodated into the PRV genome while it doesn’t influent the virus replication? Please be specific.

Response:

We thank the reviewer for this suggestion. The manuscript has been checked and revised. We have emphasized this point in the revised version in Line 35.

  1. It’s better to make a table to summarize the application of different strategies of rPRV generation and compare the advantages and disadvantages of these strategies.Response:

We thank the reviewer for this suggestion. We have made a table to summarize the application of different strategies of rPRV generation and compare the advantages and disadvantages of these strategies.

  1. For the principle behind the recombinant PRV. Whether the size of the foreign gene would impact the genome stability? Whether the insertion sites and size of the foreign gene should be considered together to construct the rPRV? If so, please discuss the relationship among the insertion site, the size of the foreign gene, and the genome stability.

Response:

We thank the reviewer for this comment. The insertion sites and size of the foreign gene should be considered together to construct the rPRV. We have point out this clearly in the revised manuscript from Line 218-227.

  1. Table 1 provides a good summary of the PRV-based vaccine. To provide a compressive overview of vaccines, the authors could consider including the immunization dose, route, animal model, and efficacy of these vaccines, such as immunogenicity and protectivity.

Response:

We thank the reviewer for this constructive comment. As you suggested we have added the information of immunization dose, route, animal model, and efficacy of these vaccines in the Table 2.

  1. Line 243, there is a grammar mistake.

Response:

We thank the reviewer for pointing out this mistake. We agree with this comment and have revised this sentence from Line 276-278.

Reviewer 2 Report

General comments:

This review aims to describe the biology of PRV, recombinant PRV construction methodology, the technology platform for efficiently constructing recombinant PRV, and the applications of recombinant PRV in vaccine development. In general, this is an interesting review highlighting the latest information on PRV usage in vaccine development against swine infectious diseases, which will offer novel perspectives for advancing preventive medicine through vaccinology. However, there are several issues that need to be addressed. These points are summarized below.

1. The title of this manuscript is a little confusing. It could be changed to read: Recombinant pseudorabies virus usage in vaccine development against swine infectious disease.

2. Line 29, “the Herpesviridae family” should be “the Herpesviridae family”.

3. The content of paragraph 2 (Lines 38-54) does not make sense. It should be deleted.

4. In section 1.1, the corresponding content is too simple. The basic principle of homologous recombination needs to be explained.

5. Line 153, “Promoter affect foreign gene expression” should be “Promoter affects foreign gene expression”.

6. Line 291, “Shanyua Zhushould be Shanyuan Zhu”?

7. The format of the reference is inconsistent and needs to be modified to be consistent.

Author Response

Revision Notes

We are grateful to the editor and reviewers for their critical comments and valuable suggestions, which have helped us to improve our manuscript. We have revised the manuscript, and would like to re-submit it for your consideration. As outlined in the following responses, we have taken all these comments and suggestions into account in the revised manuscript. The new text is highlighted in red color in the revised manuscript. We point by point responses to the referee’s comments are listed below. We hope that the revised manuscript is acceptable for publication.

Responses to the comments of Reviwer 2

This review aims to describe the biology of PRV, recombinant PRV construction methodology, the technology platform for efficiently constructing recombinant PRV, and the applications of recombinant PRV in vaccine development. In general, this is an interesting review highlighting the latest information on PRV usage in vaccine development against swine infectious diseases, which will offer novel perspectives for advancing preventive medicine through vaccinology. However, there are several issues that need to be addressed. These points are summarized below. 

Response:

We appreciate the reviewer’s comment. As recommended, we have taken these comments and suggestions into account in the revised manuscript.

  1. The title of this manuscript is a little confusing. It could be changed to read: Recombinant pseudorabies virus usage in vaccine development against swine infectious disease.

Response:

We thank the reviewer for this constructive comment. As you suggested we have changed the title.

  1. Line 29, “the Herpesviridae family” should be “the Herpesviridae family”.

Response:

We thank the reviewer for pointing out this mistake. We have revised it in the new version of the manuscript.

  1. The content of paragraph 2 (Lines 38-54) does not make sense. It should be deleted. Response: 

    We thank the reviewer for this comment. We have revised the introduction of the manuscript and added a section of “The biological characteristics of PRV”.

  1. In section 1.1, the corresponding content is too simple. The basic principle of homologous recombination needs to be explained.

Response:

We thank the reviewer for this constructive comment. As you suggested we have explained the basic principle of homologous recombination from Line 81-91.

  1. Line 153, “Promoter affect foreign gene expression” should be “Promoter affects foreign gene expression”.

Response:

We thank the reviewer for pointing out this mistake. We have revised “Promoter affect foreign gene expression” as “Promoter affects foreign gene expression”.

  1. Line 291, “Shanyua Zhu” should be “Shanyuan Zhu”?

Response:

We thank the reviewer for pointing out this mistake. We have revised this mistake in the revised manuscript.

  1. The format of the reference is inconsistent and needs to be modified to be consistent.

Response:

We thank the reviewer for this advice. As recommended, we have checked the reference.

Reviewer 3 Report

The manuscript summarises published studies of pseudorabies virus use in the generation of live-virus vectored vaccines against diseases of pigs.

Overall, seventeen specific studies are reviewed. The approaches to constructing recombinants are reviewed, including the methods, choice of insertion sites, vector choice, and the potential genes to be expressed.

The review appears to be comprehensive, and is informative, with the potential applications summarised in the conclusion.

In the introduction, the current nomenclature of PRV should be updated, using a current reference (2019 onwards).

Also in the introduction, reference 11 does not appear to be appropriate for the point made in that sentence.

Author Response

Revision Notes

We are grateful to the editor and reviewers for their critical comments and valuable suggestions, which have helped us to improve our manuscript. We have revised the manuscript, and would like to re-submit it for your consideration. As outlined in the following responses, we have taken all these comments and suggestions into account in the revised manuscript. The new text is highlighted in red color in the revised manuscript. We point by point responses to the referee’s comments are listed below. We hope that the revised manuscript is acceptable for publication.

Responses to the comments of Reviwer 3

The manuscript summarises published studies of pseudorabies virus use in the generation of live-virus vectored vaccines against diseases of pigs. Overall, seventeen specific studies are reviewed. The approaches to constructing recombinants are reviewed, including the methods, choice of insertion sites, vector choice, and the potential genes to be expressed. The review appears to be comprehensive, and is informative, with the potential applications summarised in the conclusion. In the introduction, the current nomenclature of PRV should be updated, using a current reference (2019 onwards). Also in the introduction, reference 11 does not appear to be appropriate for the point made in that sentence.

Response:

We sincerely thank the reviewer for his/her constructive and positive comments, which have helped us to improve our manuscript. As outlined in the following responses, we have taken these comments and suggestions into account in the revised manuscript. We have updated the current nomenclature of PRV; and we also deleted reference 11 in that sentence.
